# Subdomains of the *Helicobacter pylori* Cag T4SS outer membrane core complex exhibit structural independence

Jacquelyn R Roberts[1,2,*], Sirena C Tran[3,4,*], Arwen E Frick-Cheng[1], Kaeli N Bryant[4], Chiamaka D Okoye[4], W Hayes McDonald[5,8], Timothy L Cover[3,4,6], Melanie D Ohi[1,7]

The *Helicobacter pylori* Cag type IV secretion system (Cag T4SS) has an important role in the pathogenesis of gastric cancer. The Cag T4SS outer membrane core complex (OMCC) is organized into three regions: a 14-fold symmetric outer membrane cap (OMC) composed of CagY, CagX, CagT, CagM, and Cag3; a 17-fold symmetric periplasmic ring (PR) composed of CagY and CagX; and a stalk with unknown composition. We investigated how CagT, CagM, and a conserved antenna projection (AP) region of CagY contribute to the structural organization of the OMCC. Single-particle cryo-EM analyses showed that complexes purified from Δ*cagT* or Δ*cagM* mutants no longer had organized OMCs, but the PRs remained structured. OMCCs purified from a CagY antenna projection mutant (CagYΔAP) were structurally similar to WT OMCCs, except for the absence of the *α*-helical antenna projection. These results indicate that CagY and CagX are sufficient for maintaining a stable PR, but the organization of the OMC requires CagY, CagX, CagM, and CagT. Our results highlight an unexpected structural independence of two major subdomains of the Cag T4SS OMCC.

## Introduction

*Helicobacter pylori* is a Gram-negative bacterium infecting over half of the world's population (Hooi et al, 2017). *H. pylori* colonization of the stomach results in gastric inflammation (gastritis) and an increased risk for the development of gastric cancer, peptic ulcer disease, and mucosa-associated lymphoid tissue lymphoma (Atherton, 2006; Kusters et al, 2006; Malfertheiner et al, 2023). *H. pylori* strains harboring the *cag* pathogenicity island (*cag* PAI), a 40-kb chromosomal region that encodes a type IV secretion system (T4SS) and the secreted effector protein CagA, are more frequently associated with gastric cancer and peptic ulcer disease, compared with *cag* PAI–negative strains (Blaser et al, 1995; Parsonnet et al, 1997; Nomura et al, 2002; Plummer et al, 2007; Cover, 2016; Tran et al, 2024).

T4SSs are used by many different species of bacteria to transport DNA, proteins, and other substrates across the bacterial envelope (Costa et al, 2024). The components of T4SSs in Gram-negative bacteria are organized, at a minimum, into an outer membrane core complex (OMCC) and an inner membrane complex (Macé et al, 2022; Sheedlo et al, 2022; Costa et al, 2024). The OMCC is positioned in the periplasm between the bacterial outer and inner membranes (OM and IM, respectively). The *Agrobacterium tumefaciens* VirB/VirD4 T4SS, several conjugation systems (e.g., pKM101 and R388), and the *Xanthomonas citri* T4SS are considered prototypical or "minimized" T4SSs (Costa et al, 2021, 2024; Sheedlo et al, 2022). The OMCCs of prototype T4SSs consist of VirB7, VirB9, and VirB10 (Costa et al, 2021, 2024; Sheedlo et al, 2022). Expanded T4SSs are more complex than prototype (minimized) T4SSs, and typically have OMCCs composed of homologs of VirB7, VirB9, and VirB10, as well as additional species-specific components. Examples of bacteria that contain expanded T4SSs include *H. pylori*, *Legionella pneumophila*, and *Coxiella burnetii* (Costa et al, 2021, 2024; Sheedlo et al, 2022).

The *H. pylori* Cag T4SS OMCC consists of CagX, CagY, CagT, CagM, and Cag3 (Frick-Cheng et al, 2016; Chung et al, 2019; Sheedlo et al, 2020, 2022). Although CagT, CagX, and CagY share regions of structural homology with VirB7, VirB9, and VirB10, respectively, CagM and Cag3 are species-specific proteins found only in *H. pylori* (Cendron & Zanotti, 2011; Fischer, 2011; Frick-Cheng et al, 2016; Chung et al, 2019; Sheedlo et al, 2020). The *H. pylori* Cag T4SS OMCC is organized into three subassemblies: the outer membrane cap (OMC), the periplasmic ring (PR), and the stalk. Within the *H. pylori* Cag T4SS OMCC, there is a mismatch in symmetry elements between the 14-fold symmetric OMC and the 17-fold symmetric PR (Fig 1) (Chung et al, 2019; Sheedlo et al, 2020). Symmetry mismatches,

---

[1]Life Sciences Institute, University of Michigan, Ann Arbor, MI, USA    [2]Department of Biological Chemistry, University of Michigan, Ann Arbor, MI, USA    [3]Department of Medicine, Vanderbilt University School of Medicine, Nashville, TN, USA    [4]Department of Pathology, Microbiology and Immunology, Vanderbilt University School of Medicine, Nashville, TN, USA    [5]Proteomics Laboratory, Mass Spectrometry Research Center, Vanderbilt University School of Medicine, Nashville, TN, USA    [6]Veterans Affairs Tennessee Valley Healthcare System, Nashville, TN, USA    [7]Department of Cell and Developmental Biology, University of Michigan, Ann Arbor, MI, USA    [8]Department of Biochemistry, Vanderbilt University, Nashville, TN USA

Correspondence: timothy.l.cover@vumc.org; mohi@umich.edu
*Jacquelyn R Roberts and Sirena C Tran contributed equally to this work

---

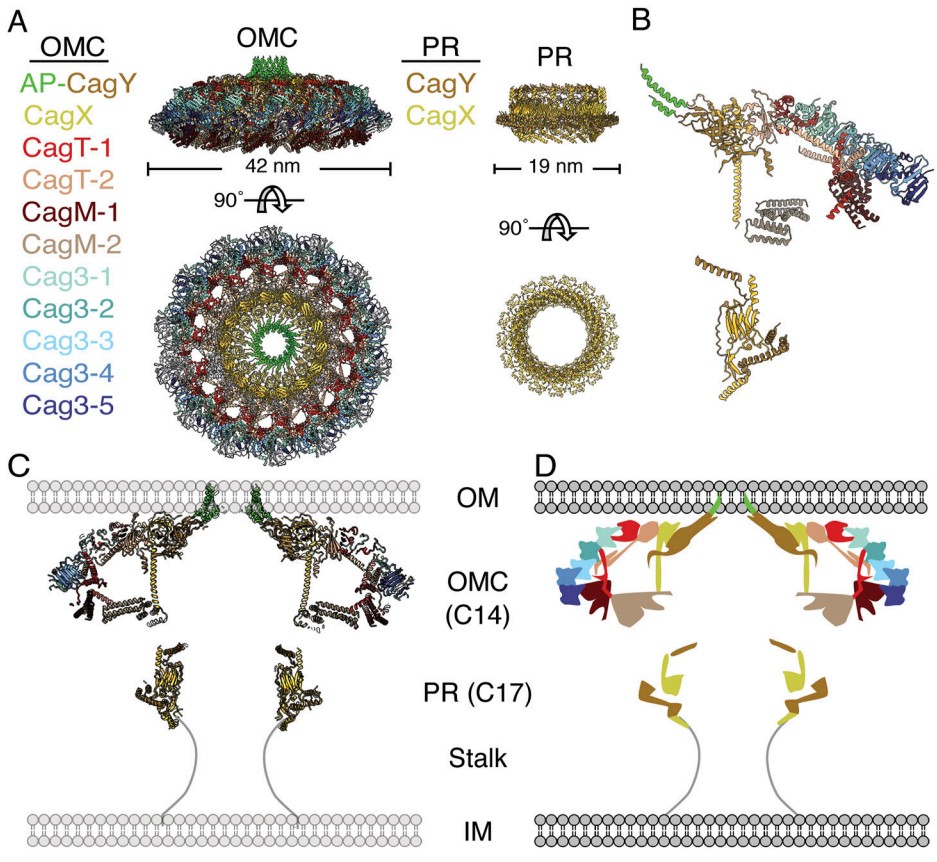

**Figure 1.  Structural organization of the *H. pylori* Cag T4SS outer membrane core complex (OMCC).**
**(A)** Molecular models of the OMC and PR rotated 90° around the x-axis (PDB: 6X6S and 6X6J). The outer membrane cap (OMC) has 14-fold symmetry, and the periplasmic ring (PR) has 17-fold symmetry. Colors for each protein in the OMC and PR are maintained throughout all panels (A, B, C, D). **(B)** One asymmetric unit of the OMC consists of CagY:CagX:CagT:CagM:Cag3 (1:1:2:2:5 ratio), and the PR consists of CagX:CagY (1:1 ratio). **(C)** Cross-section of molecular models of the OMC and PR in relation to the bacterial outer and inner membranes (OM and IM, respectively). There is no high-resolution structure of the stalk region of the OMCC (position indicated by thin black lines). **(C, D)** Cartoon depiction of the cross-section of the Cag T4SS OMCC shown in panel (C).

although with different ratios, have also been detected in OMCCs from other T4SSs, including the *L. pneumophila* Dot/Icm T4SS (Durie et al, 2020; Sheedlo et al, 2021), the F-type T4SS (Amin et al, 2021; Liu et al, 2022), and the R388 conjugation system (Macé et al, 2022).

The OMC of the *H. pylori* Cag T4SS OMCC is a ~42-nm-diameter "cap"-shaped subassembly that abuts the bacterial OM (Chang et al, 2018; Hu et al, 2019) (Fig 1). Structural analysis of the Cag T4SS OMCC by single-particle cryo-electron microscopy (cryo-EM) revealed the structural organization of CagX, CagY, CagT, CagM, and Cag3 and showed that within each asymmetric unit (ASU) of the OMC, these components are present in a stoichiometric ratio of 1:1:2:2:5 (CagX:CagY:CagT:CagM:Cag3) (Fig 1A and B) (Sheedlo et al, 2020, 2022). Structural analysis highlighted the numerous and intricate protein–protein interactions among components of the OMC.

The PR of the Cag T4SS sits directly below the OMC in the periplasm (Fig 1). This part of the OMCC contains CagY and CagX in a stoichiometric ratio of 1:1 (Fig 1A). Although the PR has a different symmetry than the OMC, it is physically connected to the OMC by both CagY and CagX, which bridge the symmetry mismatch in a way that has not been fully defined (Chung et al, 2019; Sheedlo et al, 2020). The stalk, spanning the distance between the PR and the IM, is the least characterized region of the OMCC because of the very low-resolution map available for this part of the structure (Chung et al, 2019) (Fig 1C and D). The composition and potential symmetry of the stalk have not been determined.

The portion of CagY localized to the OMC contains helix–loop–helix elements (residues 1,793–1,863) that are designated as the antenna projection (AP) (Sheedlo et al, 2020; Tran et al, 2023). The AP region of CagY is structurally similar to corresponding α-helical domains of VirB10 in conjugation systems and other prototype T4SSs (Chandran et al, 2009; Fronzes et al, 2009; Jakubowski et al, 2009; Banta et al, 2011; Sgro et al, 2018; Chung et al, 2019; Darbari et al, 2020; Sheedlo et al, 2020; Costa et al, 2021; Tran et al, 2023). Based on cryo-electron tomography (cryo-ET) analysis of intact bacteria, the part of the OMCC containing the CagY AP is localized in close proximity to the OM, and single-particle cryo-EM analysis of purified complexes showed that the 14 CagY AP elements organize into an α-helical bundle (Chang et al, 2018; Chung et al, 2019; Hu et al, 2019). The CagY antenna region is predicted to form an outer membrane pore, similar to corresponding VirB10 domains in other T4SSs (Chandran et al, 2009; Banta et al, 2011; Sgro et al, 2018; Chung et al, 2019; Darbari et al, 2020; Tran et al, 2023).

All five protein components of the Cag T4SS OMCC are required for Cag T4SS activity (Fischer et al, 2001; Johnson et al, 2014; Frick-Cheng et al, 2016). Single-particle cryo-EM analysis showed the structures and locations of CagX, CagY, CagT, CagM, and Cag3 in the OMCC and mapped the numerous protein–protein interactions among the components (Chung et al, 2019; Sheedlo et al, 2020). Previous analyses suggested that OMCCs do not form in Δ*cagX* or Δ*cagY* mutant strains (Frick-Cheng et al, 2016; Hu et al, 2019). Analysis of a Δ*cag3* mutant strain showed that a stable OMCC could

form in the absence of Cag3 (Sheedlo et al, 2020), but the roles of the other OMCC components (CagT and CagM) in the structural stability of the complex have not been carefully analyzed.

In this study, we expand our understanding of the roles of CagT, CagM, and the AP region of CagY in the organization of the *H. pylori* Cag T4SS OMCC. We use single-particle cryo-EM analysis and mass spectrometry to investigate the structure and composition of OMCCs purified from Δ*cagT* and Δ*cagM* mutant strains, and we analyze the structure of OMCCs purified from an *H. pylori* strain lacking the CagY AP (CagYΔAP). We show that OMCCs purified from the CagYΔAP mutant have structures similar to those of WT OMCCs, except for the absence of the CagY AP region. Complexes purified from the Δ*cagT* mutant lack CagT and Cag3, leading to destabilization of the OMC, but maintain a structured PR. Similarly, complexes purified from the Δ*cagM* mutant lack CagM, CagT, and Cag3 and no longer have a structured OMC, but retain a structured PR. These data indicate that CagX and CagY are sufficient for maintaining a stable PR, but the structural organization of the OMC requires four proteins (CagX, CagY, CagT, and CagM). These results highlight the numerous protein–protein interactions required for OMC organization, the unexpected structural independence of the OMC and PR subdomains, and the finding that the organization of the OMCC does not require the CagY AP.

# Results

## CagT is essential for OMC structural stability and non-essential for PR stability

First, we sought to determine the structural organization of the OMCC in the absence of the CagT. CagT is a lipoprotein that has limited sequence similarity to VirB7 homologs, but it shares structural relatedness to *X. citri* VirB7 within the N-terminal region (Fischer, 2011; Backert et al, 2017; Sgro et al, 2018; Chung et al, 2019; McClain et al, 2020). Two copies of CagT are present in each ASU of the OMC (Sheedlo et al, 2020) (Fig 1A and B). Both copies are localized in the outer layer of this subassembly, and the cysteine residue predicted to be lipidated is positioned to interact with the bacterial outer membrane (Chung et al, 2019; Sheedlo et al, 2020). Studies of a mutant strain engineered to produce CagT with an altered lipobox indicated that CagT lipidation is essential for CagT stability and Cag T4SS activity (McClain et al, 2020). A Δ*cagT* deletion mutant is defective in both the translocation of CagA into gastric epithelial cells and the induction of IL-8 in AGS gastric epithelial cells (Fischer et al, 2001; Johnson et al, 2014; Frick-Cheng et al, 2016), indicating that CagT is required for Cag T4SS function. Immunoblot analysis of Cag T4SS complexes isolated from the Δ*cagT* mutant showed that the preparations did not contain CagT or Cag3, but CagY, CagX, and CagM were still detected (Frick-Cheng et al, 2016). Cryo-ET analysis of OMCCs in the Δ*cagT* bacteria lacked peripheral density, and fewer T4SS complexes were detected in the Δ*cagT* mutant than in WT bacteria (Hu et al, 2019). Negative stain EM analysis of Cag T4SS complexes purified from the Δ*cagT* mutant showed no structured OMCCs but did show some thin rings with a ~19-nm diameter (Frick-Cheng et al, 2016). However, the lack of a

**Table 1. LC-MS/MS analysis of Cag T4SS complexes isolated from mutant strains[a].**

| Identified proteins | WT[b] | Δ*cagT* | WT[b] | Δ*cagM* |
|---|---|---|---|---|
| | Assigned spectral counts | | | |
| CagA | 715 | 718 | 890 | 769 |
| CagF | 124 | 149 | 79 | 76 |
| CagX | 175 | 199 | 250 | 101 |
| CagM | 97 | 82 | 106 | 0 |
| Cag3 | 244 | 2 | 77 | 2 |
| CagT | 130 | 0 | 102 | 0 |
| CagY | 560 | 752 | 581 | 264 |
| Other Cag proteins | 52 | 49 | 104 | 27 |
| No. of Cag spectra | 2,097 | 1,951 | 2,112 | 1,237 |
| No. of non-Cag spectra | 980 | 565 | 427 | 392 |
| Total spectral counts | 3,077 | 2,516 | 2,539 | 1,629 |

[a]Each of the strains contained sequences encoding HA-CagF. HA-CagF was purified as described in the Materials and Methods section, and the composition of the resulting preparations was analyzed by mass spectrometry. The list of *H. pylori* proteins identified was filtered using a 1% peptide false discovery rate and a minimum of two distinct peptides per protein. The table shows the numbers of detected peptides matched to the indicated proteins.
[b]Results are shown for WT OMCCs analyzed in two separate experiments.

high-resolution OMCC structure available at the time of the previous work limited any detailed conclusions that could be made about the role of CagT in the overall organization of OMCC, other than concluding that it was important for complex stability. Now, with the availability of a 3.4 Å resolution structure and the molecular model of the WT Cag T4SS OMCC (Fig 1) (Chung et al, 2019; Sheedlo et al, 2020), we decided to more closely examine how CagT, a conserved T4SS component, contributes to the overall structural organization of the OMCC.

Using mass spectrometry, negative stain EM, and single-particle cryo-EM analyses, we analyzed the composition and structure of Cag T4SS complexes purified from the Δ*cagT* mutant. Liquid chromatography–tandem mass spectrometry (LC-MS/MS) analysis of T4SS complexes purified from the Δ*cagT* mutant showed that the samples contained CagX, CagY, and CagM, but lacked Cag3 and CagT (Table 1). These results concur with previous immunoblot results (Frick-Cheng et al, 2016). As before, negative stain EM showed that complexes isolated from the Δ*cagT* mutant differed markedly from WT OMCCs (Fig 2A), with only ~19-nm-wide thin rings visible in the images of T4SS complexes purified from the Δ*cagT* mutant (Fig 2B). However, now with a better understanding of the overall Cag T4SS OMCC structure, we recognized that these rings have a size and appearance similar to the PR of the OMCC.

To examine this more closely, we collected a cryo-EM dataset of complexes purified from the Δ*cagT* mutant and used 2D and 3D single-particle analysis to characterize the complexes (Fig S1A and B, Table S1). The "*en face*" views of 2D classes of mutant particles contain none of the secondary structural features of the OMC that usually dominate this view of WT OMCCs (Fig 2C and D), indicating that this part of the complex is no longer stably organized in the

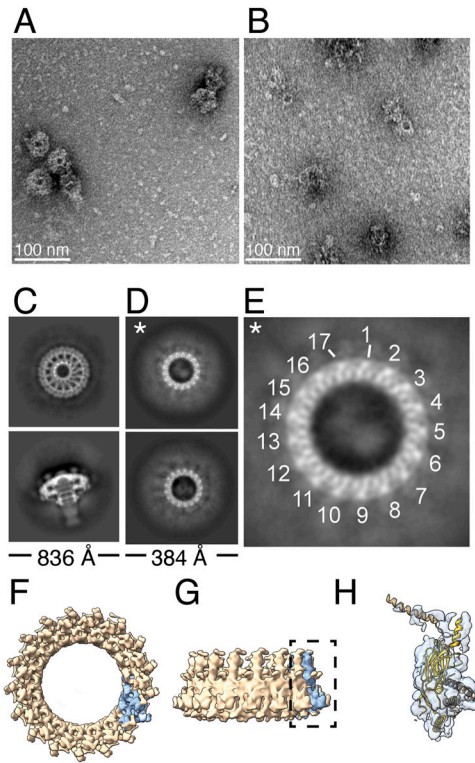

**Figure 2.  Structural analysis of the *H. pylori* Cag T4SS outer membrane core complex (OMCC) purified from the Δ*cagT* mutant.**
**(A, B)** Representative negative stain images of Cag T4SS complexes purified from WT (A) or Δ*cagT* (B) mutants. Scale bars, 100 nm. **(C)** Representative cryo-EM 2D class of averages of an "*en face*" and side view of WT outer membrane core complexes. Side length of box, 836 Å. **(D)** Representative cryo-EM 2D class averages of T4SS complexes purified from the Δ*cagT* mutant. Side length of box, 384 Å. * represents class shown enlarged and cropped in panel (E). **(E)** Enlarged 2D class with 17-fold symmetry marked. **(F, G)** 8 Å resolution 3D EM density map of the PR with 17-fold applied symmetry. Structure in panel (F) is shown from the perspective of the outer-to-inner membrane view, and the structure in panel (G) is rotated 90° on the x-axis. One asymmetric unit of the PR EM density map, which contains CagX and CagY, is colored in blue. **(H)** Enlarged view of one asymmetric unit of the Δ*cagT* PR. EM density is transparent blue. Molecular models of the regions of CagY (brown, residues 1,469–1,603) and CagX (yellow, residues 32–130, 261–323) found in the WT PR (PDB: 6X6J) have been placed into the EM density.

absence of CagT. However, the averages did show the presence of a structured ring, ~19 nm in diameter with 17-fold symmetry (Fig 2E). This is the same width and symmetry as the PR in WT OMCCs (Fig 1A). Thus, 2D analysis of the cryo-EM images indicates that T4SS complexes lacking CagT no longer have a structured OMC but appear to retain an organized PR.

3D reconstructions of complexes purified from the Δ*cagT* mutant confirmed the 2D analysis. Although there is amorphous density in the 3D reconstruction that can be attributed to the OMC, the density is extremely noisy with no apparent secondary structural features or symmetry (Fig S1). Although the mass spectrometry analysis detected CagM, CagY, and CagX in the samples, including CagY and CagX peptides from regions localized in the OMC, neither the 2D averages nor the 3D reconstruction showed any defined structural features that can be attributed to these proteins in the OMC. However, the Δ*cagT* complexes still have a structurally defined PR. A

3D structure of the PR with 17-fold symmetry applied reached 8.0 Å resolution (Fig S2A–C) and had secondary structural features that made it possible to place molecular models for CagY (residues 1,469–1,603) and CagX (residues 32–130, 261–323) built from the WT PR EM density map into the Δ*cagT* PR density map (Fig 2F–H). Although there are some minor differences between the high-resolution WT PR model and the Δ*cagT* PR map (Fig S3A), likely because of the heterogeneity in the Δ*cagT* particles, the overall structural integrity of the PR is preserved.

Overall, these analyses show that CagT is required for the stable association of Cag3 with the OMCC and is essential for the structural organization of CagM, CagX, and CagY within the OMC. However, even in the absence of an organized OMC, the overall structural organization of the PR is maintained.

## Loss of the CagY AP region does not alter the structural organization of the OMCC

The importance of CagT in the OMC structure might be explained solely by the central position of the 28 copies of CagT found in each OMC (Fig 1) (Sheedlo et al, 2020). However, CagT is also a lipoprotein, with the position of post-translational lipid moieties likely providing a mechanism to anchor the OMC to the outer membrane (McClain et al, 2020; Sheedlo et al, 2020). Thus, CagT might contribute to the OMC structure through its interactions with the outer membrane. Another Cag T4SS protein that contacts the outer membrane is CagY (Chung et al, 2019; Sheedlo et al, 2020). The CagY AP motif is highly conserved among VirB10 family members, is predicted to form a channel in the outer membrane, and is required for *H. pylori* Cag T4SS activity (Tran et al, 2023). Specifically, an *H. pylori* mutant lacking the CagY AP (residues 1,793–1,863) is defective in IL-8 induction and CagA translocation (Tran et al, 2023). VirB10 AP regions are predicted to form a channel through the outer membrane, and cryo-ET analysis of the *H. pylori* T4SS showed that the OMCC region containing the CagY AP is localized adjacent to the outer membrane (Chang et al, 2018; Hu et al, 2019). If OMC interactions with the outer membrane contribute to OMC assembly or overall stability, the CagY AP motif would be predicted to be important for OMC organization. To test this hypothesis, we investigated whether the presence or absence of the CagY AP region influences the overall structure of the OMCC.

To examine the structural contribution of the conserved AP region of CagY to OMCC organization, we used both negative stain EM and single-particle cryo-EM analyses to investigate the structural organization of Cag T4SS OMCCs purified from a CagYΔAP mutant strain. Negative stain analysis showed that OMCCs from the CagYΔAP mutant globally resembled WT OMCCs (Figs 2A and 3A). To determine whether there were more nuanced changes in the architecture of the Cag T4SS not evident by low-resolution negative stain imaging, we then examined these OMCCs using cryo-EM analysis (Fig S4A and B, Table S1). 2D analysis of particles in vitrified ice showed that the "*en face*" views of the CagYΔAP OMCCs contained clear secondary structural elements and were similar in diameter to WT OMCCs (Figs 2C and 3B); however, the innermost central density, corresponding to where 14 copies of the CagY AP region form an α-helical channel, is missing in the mutant complexes (Fig 3B). 2D classes representing side views of the mutant

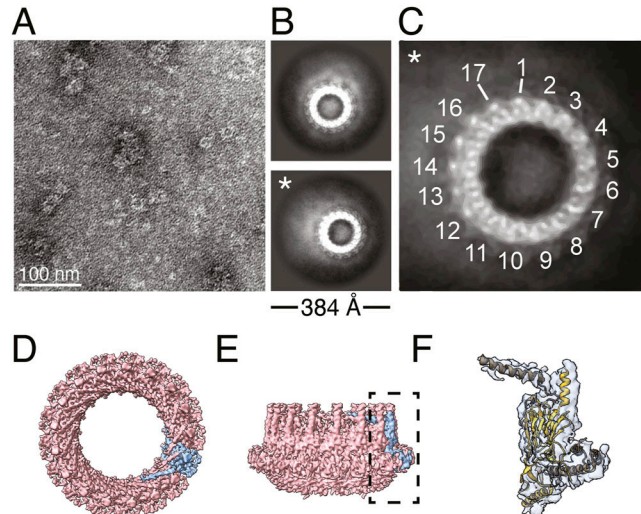

**Figure 3. Structural analysis of the *H. pylori* Cag T4SS outer membrane core complex (OMCC) purified from the CagYΔAP mutant.**
**(A)** Representative negative stain image of OMCCs purified from the CagYΔAP mutant. Scale bar, 100 nm. **(B)** Cryo-EM 2D class averages of "*en face*" and side views of OMCCs purified from the CagYΔAP mutant strain. The white arrow in one of the side-view 2D averages marks the position where the CagY AP region would be in WT complexes. Side length of box, 600 Å. **(C, D)** 3.8 Å resolution 3D EM density map of the CagYΔAP outer membrane cap (OMC) with 14-fold applied symmetry. *En face* view of the OMC in panel (C) is shown from the perspective of looking from the outside to the inside of the bacteria (i.e., outer to inner membrane), and the cross-section of the OMC map in panel (D) is rotated 90° on the x-axis. Dashed box is the region shown enlarged in panel (E). Position of CagY shown in blue, and model AP domain (missing in structure) in green. **(E)** Enlarged EM density of the position of one CagY in the CagYΔAP OMC map (transparent blue). EM density is transparent blue with the molecular model of WT CagY (from PDB: 6X6J) placed into the CagYΔAP OMC map. CagY residues 1,677–1,909 are shown in brown with the AP (residues 1,793–1,863) shown in green. **(F, G)** 6.6 Å resolution 3D EM density map of the PR with 17-fold applied symmetry. One asymmetric unit of the EM density map is colored in blue. Structure in panel (F) is shown from the perspective of an outer-to-inner membrane view, and the cross-section of the map of the PR in panel (G) is rotated 90° on the x-axis. Dashed box is the region shown in panel (H). **(H)** Enlarged EM density of one asymmetric unit of the PR (transparent blue). Molecular models of the regions of CagY (brown, residues 1,469–1,603) and CagX (yellow, residues 32–130, 261–323) found in the WT PR (PDB: 6X6J) have been placed into the CagYΔAP EM density map.

complexes showed pairs of CagYΔAP OMCCs interacting where the channel is usually located (Fig 3B). This interaction between OMCCs has not been observed in analyses of WT or Δ*cag3* OMCCs (Chung et al, 2019; Sheedlo et al, 2020). 3D single-particle cryo-EM analysis of OMCCs purified from the CagYΔAP mutant showed that although the complexes clearly lack the AP extension, the maps of the OMC and the PR (at the resolutions of 3.8 and 6.6 Å, respectively) (Fig S5A–F) are very similar to the corresponding maps of the WT OMC and PR (Figs 3C–H and S3B and C). CagY in the CagYΔAP OMC, other than missing the helix–loop–helix regions (corresponding to the deleted amino acids 1,793–1,863), adopts the same conformation as CagY in the WT OMC (Fig 3C–H). The CagYΔAP PR also has the same overall organization as the WT PR, although there are some minor differences that cannot be directly modeled at the current resolution of the map (Fig S3C). Thus, although the CagY AP region is required for the biological function of the *H. pylori* Cag T4SS (Tran et al, 2023) and likely interacts with the OM (Chang et al, 2018; Hu et al, 2019), it is not required for the overall structural stability and organization of the OMCC.

## CagM, an *H. pylori*-specific T4SS component, is required for a structured OMC but not PR stability

We next characterized the role of the *H. pylori*-specific T4SS protein CagM in the structural organization of the OMCC. CagM does not have obvious homologs in any non-*H. pylori* species (Sheedlo et al, 2020). As with CagT, there are two copies of CagM found in each ASU of the OMC (Fig 1A); however, unlike CagT, both copies of CagM are found in the inner layer, rather than the outer layer, of this sub-assembly (Fig 1) (Chung et al, 2019; Sheedlo et al, 2020). CagM is essential for Cag T4SS function, because a Δ*cagM* mutant is defective in the translocation of CagA and induction of IL-8 in AGS gastric cells (Fischer et al, 2001; Johnson et al, 2014; Frick-Cheng et al, 2016). Based on immunoblot analysis, OMCCs purified from the Δ*cagM* mutant only contain CagX and CagY, with CagM, CagT, and Cag3 not detected (Frick-Cheng et al, 2016). To carefully examine the role of this species-specific T4SS protein in the structural organization of the Cag T4SS OMCC, we purified T4SS complexes from the

**Figure 4. Structural analysis of the *H. pylori* Cag T4SS of the outer membrane core complex (OMCC) purified from the Δ*cagM* mutant.**
**(A)** Representative negative stain image of complexes purified from the Δ*cagM* mutant. Scale bar, 100 nm. **(B)** Representative cryo-EM 2D class averages of T4SS complexes purified from the Δ*cagM* mutant. * represents class shown enlarged and cropped in panel (C). Side length of box, 384 Å. **(C)** Enlarged 2D class (*) with 17-fold symmetry highlighted. **(D, E)** 8.5 Å resolution 3D EM density map of the PR with 17-fold applied symmetry. Structure in panel (D) is shown from the perspective of an outer-to-inner membrane view. One asymmetric unit (ASU) of the PR EM density map, which contains CagX and CagY, is colored in blue. Structure shown in panel (E) is rotated 90° around the x-axis in relation to structure shown in (D). Dashed box identifies ASU enlarged in panel (F). **(F)** Enlarged view of one ASU of the PR. EM density is transparent blue. Molecular models of the regions of CagY (brown, residues 1,469–1,603) and CagX (yellow, residues 32–130, 261–323) found in the WT PR (PDB: 6X6J) have been placed into the EM density.

Δ*cagM* mutant and examined their composition and structure using mass spectrometry, negative stain EM, and single-particle cryo-EM analyses.

LC-MS/MS analysis of complexes purified from the Δ*cagM* mutant showed that the preparations contained CagX and CagY, but lacked CagT, CagM, and Cag3 (Table 1), matching the previous

immunoblot analysis (Frick-Cheng et al, 2016). Examination of the mutant particles by negative stain revealed thin rings similar to complexes purified from the ΔcagT mutant (Figs 2B and 4A). We speculate that the ability to visualize these structures in the current study but not in a previous study (Frick-Cheng et al, 2016) is attributable to the use of larger volume bacterial cultures and improved purification strategies.

We collected a cryo-EM dataset and analyzed the resulting particles using 2D and 3D single-particle cryo-EM approaches (Fig S6A and B, Table S1). In the 2D averages, the ΔcagM complexes have no structured OMC (Fig 4B), in contrast to the well-defined OMC visible in WT OMCCs (Fig 2C); however, there is density for the PR with clear secondary structural features and 17-fold symmetry (Fig 4B and C). 3D reconstruction of these particles confirmed the observations based on analysis of the 2D averages and showed there was no structured density for the OMC (Fig S6A). However, the ΔcagM complexes still have a structurally defined PR. The 3D reconstruction of the PR from the ΔcagM mutant reached 8.5 Å with 17-fold symmetry imposed (Fig S7A–C), and secondary structural features expected at this resolution were visible in the map (Fig 4D–F). We can place molecular models of the regions of CagY (residues 1,469–1,603) and CagX (residues 32–130, 261–323) found in the PR directly into this density (Fig 4F), showing that these parts of CagX and CagY remain structured in the ΔcagM background. Although there are some subtle differences between the 3D structure of the ΔcagM PR and WT PR (Fig S3D), this analysis shows that the ΔcagM PR, even without a structured OMC, retains its overall structural organization. Therefore, these results indicate that CagM is required for CagT and Cag3 association with the OMCC, is required for the structural organization of the OMC, and is not required for maintaining the structural organization of the PR.

## Discussion

In this study, we investigated the roles of individual *H. pylori* Cag T4SS OMCC proteins in the stability of the OMCC. Previous studies of *H. pylori* deletion mutants lacking OMCC components, using low-resolution negative stain EM and cryo-ET approaches, provided preliminary evidence that there were differences in how the individual proteins contribute to the overall OMCC structural organization and stability. OMCCs could not be purified at all from ΔcagY and ΔcagX mutants, and the Cag T4SS was not visible by cryo-ET analysis in bacteria lacking either CagY or CagX (Frick-Cheng et al, 2016; Hu et al, 2019). In contrast, structured OMCCs, composed of CagY, CagX, CagM, and CagT, were isolated from a Δcag3 mutant (Frick-Cheng et al, 2016; Sheedlo et al, 2020). These results suggested a model in which evolutionarily conserved T4SS components (such as CagX and CagY) might play a more important role in the overall stability and structural organization of the Cag T4SS OMCC than species-specific components (such as Cag3). Our careful examination in the current study of T4SS complexes purified from strains lacking either CagT, a Cag T4SS component whose N-terminus is structurally related to the corresponding region of *X. citri* VirB7 (Sgro et al, 2018; Sheedlo et al, 2020), or CagM, an *H. pylori*-specific protein, shows that this model is too simplistic. Although

OMCCs isolated from ΔcagT and ΔcagM mutants had structured PRs, neither had structured OMCs. Thus, whether a T4SS protein is conserved or species-specific does not, de facto, provide information about its role in the T4SS structural organization or stability. Instead, each protein's contribution to OMCC organization depends more on its location within the structure than whether it is conserved across species (Fig 5).

The characteristics of T4SS complexes purified from *H. pylori* mutant strains lacking individual T4SS components are consistent with what might be predicted by examining the structure of the WT Cag T4SS OMCC. The cryo-EM maps of the WT *H. pylori* Cag T4SS OMCC showed that protein–protein interactions among the five protein components, especially in the OMC, create an intricate network of interdependence. These include interactions between duplicate copies of CagT, interactions between duplicate copies of CagM, interactions among the multiple copies of Cag3, and CagT-Cag3, CagT-CagM, CagT-CagY, Cag3-CagM, CagM-CagX, CagT-CagY, and CagX-CagY interactions (Sheedlo et al, 2020). The requirement of CagY and CagX for the formation of a stable OMCC (Frick-Cheng et al, 2016) is consistent with the presence of both proteins in the PR, a role of both proteins in physically traversing the symmetry mismatch between the PR and OMC, and the presence of both proteins in a central location within the OMC (Fig 5A). The localization of CagT and CagM within the Cag T4SS OMCC structure is also consistent with the roles of these proteins in the formation of a stable OMC. CagT and CagM make up either the outer or inner "spokes" of the OMC that help connect CagX and CagY in the center of the complex with Cag3 located at the periphery (Fig 5A) (Chung et al, 2019; Sheedlo et al, 2020). In the absence of either CagT or CagM, the structural stability of these spokes breaks down, disrupting the organization of the OMC (Fig 5D and E). In contrast, the multiple copies of Cag3 are localized at the periphery of the OMC (Chung et al, 2019; Sheedlo et al, 2020) and are not required for stable interactions between CagM, CagT, CagX, and CagY. Thus, in the absence of Cag3, the OMC has a smaller diameter but otherwise remains structured (Fig 5C) (Sheedlo et al, 2020).

Although the structure of the Cag T4SS explains the roles of CagX, CagY, CagM, CagT, and Cag3 in OMC organization, it was not obvious how the loss of a structured OMC would affect the integrity of the PR. Parts of CagX and CagY make important structural contributions to the PR and the OMC, and CagX and CagY bridge the symmetry mismatch between these subassemblies (Sheedlo et al, 2020). Because folded domains of CagX and CagY are found in both the OMC and the PR, one model predicts that the loss of an organized OMC would also have a deleterious effect on the PR. However, our analysis of OMCCs purified from ΔcagM and ΔcagT mutants unexpectedly showed that the PR remains structurally intact with 17-fold symmetry even when the OMC is not structured (Fig 5D and E). These results provide experimental evidence for the existence of oligomerization domains within two different regions of CagX and CagY (i.e., domains of these proteins localized to either the OMC or the PR).

The PR is composed of only CagX and CagY (Sheedlo et al, 2020). In the PR, 17 copies of the N-terminus of CagX (residues 32–323) and a segment of CagY (residues 1,469–1,603) interact to form a ~19-nm ring structure. In the OMC, there are 14 copies of the C-terminal region of CagX (residues 349–515) and CagY (residues 1,677–1,910)

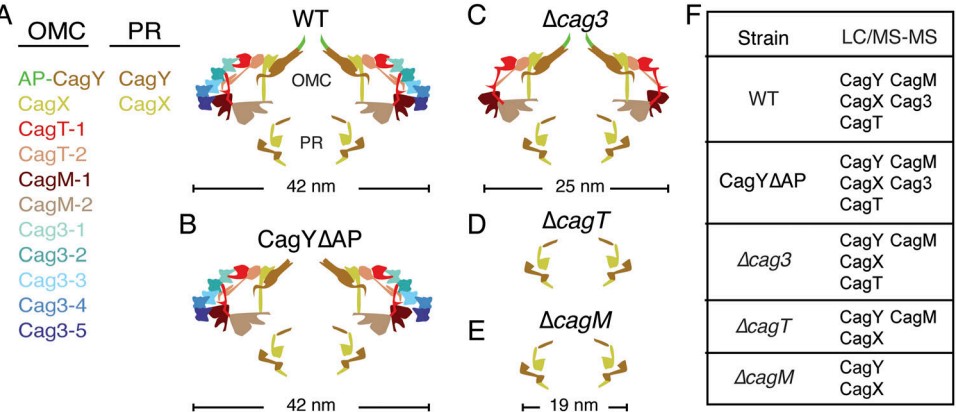

**Figure 5. Structural organization and composition of Cag T4SS outer membrane core complexes (OMCCs) purified from WT and mutant *H. pylori* strains.**
**(A, B, C, D, E)** Cartoon depictions of the Cag T4SS outer membrane core complex comparing the structural organization of complexes purified from WT strain (A), and CagYΔAP (B), Δ*cag3* (C), Δ*cagT* (D), and Δ*cagM* (E) mutants. Colors of proteins depicted in panel (A) are maintained throughout panels (B, C, D, E). **(F)** Summary of T4SS proteins detected in purified T4SS preparations by LC-MS/MS analysis. Although CagM was present in preparations from the Δ*cagT* mutant, there was no structured density for this protein seen by EM analysis.

(Sheedlo et al, 2020). Although the precise molecular details for how CagX and CagY bridge the symmetry mismatch between the PR and OMC are not known, a low-resolution non-symmetrized density map determined using a focused refinement strategy provided insight about the region of CagX that connects the subassemblies (Sheedlo et al, 2020). Although the resolution in this region was not adequate to build a molecular model, it clearly showed α-helical tubes extending from 14 of the 17 CagX densities in the PR to where CagX is localized in the OMC (Sheedlo et al, 2020). Thus, a long α-helix of CagX bridges the PR and OMC. There was no density for CagY observed in the region of the non-symmetrized structure connecting the OMC with the PR, leading to a proposed model where a stretch of CagY missing in the symmetrized maps of the PR and OMC (residues 1,604–1,676) connects these subassemblies via a flexible loop (Sheedlo et al, 2020). Our finding that the PR remains structured even in the absence of an organized OMC is consistent with a model in which the CagX and CagY regions bridging the PR and OMC are structurally flexible.

The results of the current study indicate that a Δ*cagT* mutant can assemble a stable PR, but the OMC portion of the OMCC is disorganized (Fig 5D). The simplest explanation is that the 28 copies of CagT in the OMCC are essential for the structural stability of the OMC (Fig 5A). In addition, CagT is a lipoprotein, and the amino acids predicted to be post-translationally lipidated are located at sites in the CagT structure where they could interact with the OM (Sheedlo et al, 2020). Therefore, another possible explanation for the loss of OMC stability in the Δ*cagT* background could be the disruption of important OMC interactions with the OM. An additional region of the OMCC that interacts with the outer membrane is the CagY AP, which is predicted to form an outer membrane channel (Tran et al, 2023). For this reason, we hypothesized that the CagY AP might also be required for OMC formation. To our surprise, analysis of OMCCs from a CagYΔAP mutant revealed that these complexes had structures similar to WT OMCCs (Fig 5B), except for the absence of densities corresponding to the 14 CagY AP domains in the CagYΔAP complexes. The CagYΔAP PR has some subtle differences when compared to the WT PR, suggesting that changes in the CagY AP might influence the structure of CagY in the PR. However, our current studies do not provide any mechanistic insight into how these changes would be propagated. Most importantly, these studies indicate that the CagY AP is required for Cag T4SS function (Tran et al, 2023) but is not required for OMCC assembly.

In future experiments, it will be important to determine whether the results observed in the current study extend to T4SSs in other bacterial species. For example, the *Legionella* Dot/Icm T4SS OMCC contains two well-defined OMC and PR subassemblies exhibiting symmetry mismatch (Durie et al, 2020; Sheedlo et al, 2021), so it will be interesting to determine the roles of VirB7 and *Legionella*-specific OMC components in the formation of the OMC and PR in this T4SS. Similarly, it will be interesting to determine whether the I-layer in the R388 T4SS OMCC or the *X. citri* T4SS is stable in the absence of VirB7 (Sgro et al, 2018; Macé et al, 2022). Within the F-type T4SS, the OMCC is organized into a central cone (analogous to the I-layer) and an outer ring (analogous to the O-layer); therefore, VirB7 might be non-essential for the stability of the central cone in this OMCC (Amin et al, 2021; Liu et al, 2022; Kishida et al, 2024).

Finally, although these results do not provide any direct evidence for how Cag T4SSs assemble in vivo, they do support a key role of CagX and CagY in the overall T4SS structural stability. At least in the context of purified complexes, the regions of CagX and CagY found in the OMC require both CagT and CagM to maintain their structural organization. In general, our results support a model for the T4SS assembly pathway that was proposed based on cryo-ET analyses of Cag T4SSs visualized in WT, Δ*cag3*, and Δ*cagT* bacterial cells (Hu et al, 2019). In this model, CagX, CagY, and CagM first assemble into a central "cylinder" located between the inner and outer membranes that serves as the structural scaffold for the subsequent addition of CagT and Cag3 to form the OMC. Further studies will be required to elucidate the sequence of steps required for T4SS assembly in intact bacteria.

# Materials and Methods

### Bacterial strains, plasmids, and cell culture

*H. pylori* strains were cultured on trypticase soy agar plates containing 5% sheep blood in ambient air supplemented with 5% $CO_2$.

Liquid cultures of *H. pylori* were grown in Brucella broth supplemented with 10% vol/vol heat-inactivated FBS. CagYΔAP (deletion of CagY amino acids 1,793–1,863), unmarked Δ*cagT*, and unmarked Δ*cagM* mutant strains, each containing sequences encoding HA-CagF, have been described previously (Frick-Cheng et al, 2016; Tran et al, 2023).

## T4SS OMCC purification

OMCCs were purified from mutant *H. pylori* strains using a purification method targeting HA-CagF, as described previously (Frick-Cheng et al, 2016; Chung et al, 2019). In brief, strains were grown in liquid culture for about 20 h, the bacteria were pelleted at 3,300*g* for 15 min at 4°C, and the pellet was resuspended in RIPA buffer (50 mM Hepes, 100 mM NaCl, 1% NP-40, and 0.025% deoxycholate supplemented with 1 mM phenylmethylsulfonyl fluoride and protease inhibitors [Roche]) and sonicated on ice (at 25% amp, 10 s on and 10 s off, five times). The suspension was then incubated for 1 h at 4°C. The insoluble material was pelleted, the bacterial lysate was incubated with anti-HA antibodies non-covalently linked to protein G Dynabeads (Invitrogen) for 30 min, and then the complexes were eluted with HA peptide for 1 h at RT.

## LC-MS/MS analysis of Cag T4SS OMCCs

To analyze the protein content of the immunopurified samples, samples were digested and prepared for analysis using S-traps (Protifi), following the manufacturer's recommended protocol. The resulting tryptic peptides were analyzed by data-dependent LC-MS/MS (McClain et al, 2023). Briefly, peptides were autosampled onto a 200 mm by 0.1 mm (Jupiter 3 micron, 300A) self-packed analytical column coupled directly to an LTQ linear ion trap mass spectrometer (Thermo Fisher Scientific) using a nanoelectrospray source, and resolved using an aqueous-to-organic gradient. Both the intact masses (MS) and fragmentation patterns (MS/MS) of the peptides were collected in a data-dependent manner using dynamic exclusion to maximize the depth of coverage. Resulting peptide MS/MS spectral data were searched against an *H. pylori* database to which sequences of common contaminants and reversed versions of each protein had been added, using SEQUEST. Peptide spectral matches were collated, filtered, and compared using Scaffold (Proteome Software). Protein identifications required a minimum of two unique peptides per protein and were filtered to a 1% false discovery rate for peptides and a 5% false discovery rate for proteins.

## Negative stain EM sample preparation and data collection

Negative stain EM was carried out using established methods (Ohi et al, 2004). 400-mesh copper grids covered with carbon-coated collodion film (Electron Microscopy Sciences) were glow-discharged for 30 s at 5 mA in a PELCO easiGlow glow discharge unit (Ted Pella). 3.5 µl of the Cag T4SS sample (as purified) was adsorbed to the grids and incubated for 1 min at RT. The grids were then washed twice with water, negatively stained with 0.75% (wt/vol) uranyl formate solution, and blotted until dry. Negative stain images were taken using a Tecnai Spirit T12 transmission electron microscope (Thermo Fisher Scientific) operated at 120 kV and at a nominal magnification of 26,000x (2.3 Å/pixel). Images were acquired with Leginon (Suloway et al, 2005) on a 4K × 4K Rio complementary metal-oxide semiconductor camera (Gatan) at −1.5-µm defocus value.

## Cryo-EM sample preparation

Cryo-EM samples were prepared as described previously (Chung et al, 2019; Sheedlo et al, 2020). In brief, 3.5 µl of the Cag T4SS OMCC sample (as purified) was applied to a glow-discharged Quantifoil R 2/2 UT 200-mesh copper grid (Quantifoil). The sample was applied to a grid three to four times, incubated for 60 s each before blotting, and vitrified by plunge freezing in a slurry of liquid ethane using Vitrobot (Thermo Fisher Scientific) at 20°C and 100% humidity.

## Cryo-EM data collection

Images were collected on a Titan Krios electron microscope (Thermo Fisher Scientific) equipped with K3 Summit Direct Electron Detector (Gatan) operated at 300 kV with a nominal pixel size of 1.08 Å per pixel. The Bioquantum energy filter (Gatan) was inserted with a slit width of 20 eV. Micrographs were acquired using SerialEM software (Mastronarde, 2005). The electron dose totaled 60 e/$Å^2$, and the defocus range was −1 to −2.5 µm.

## Cryo-EM data analysis

cryoSPARC v.4.2.1 was used for image processing of all cryo-EM datasets (Punjani et al, 2017). Fig S1 shows data processing steps for cryo-EM analysis of T4SS complexes purified from the Δ*cagT* mutant. For analysis of the Δ*cagT* OMCC, 310,039 particles were selected by template picking in cryoSPARC. After iterative 2D classification, classes with clear secondary structural features were retained, corresponding to 8,415 particles. These particles were used in a reference-free initial 3D reconstruction (ab initio model in cryoSPARC) designating two 3D classes, no applied symmetry (C1), and a resolution range of 35–12 Å (the default cryoSPARC setting). The 3D classes resembled previous PR structures and did not have density resembling the OMC. In an effort to visualize any structured OMC density, one of the ab initio 3D volumes was used as the initial model in a non-uniform refinement that used 8,415 particles and imposed C14 or C17 symmetry. Both reconstructions contained artifacts and lacked any secondary structural features associated with either the OMC or the PR. Because there were 2D averages with 17-fold symmetry, we did another ab initio reconstruction applying C17 symmetry, designating two 3D classes, and choosing a resolution range of 20–4 Å. The best class contained 5,706 particles. These particles were used in a subsequent local refinement, using the ab initio model as the initial volume, and a solvent mask that was created from the ab initio model that was padded and dilated 10 pixels each. The rotation search extent was 10°. After local refinement, the final resolution was 8 Å (Fig S2).

Fig S4 shows data processing steps for cryo-EM analysis of OMCCs purified from the CagYΔAP mutant. For analysis of the CagYΔAP OMCC, 633,415 particles were selected by template picking in cryoSPARC. After iterative 2D classification, classes with clear

secondary structural features were retained, containing a combined 22,643 particles. These particles were subjected to an ab initio 3D reconstruction designating one 3D class not applying symmetry, and using a resolution range of 35–12 Å. This model was used as a reference for a non-uniform 3D refinement with C1 symmetry, resulting in a density map with a global resolution of 6.8 Å. For the focused refinement of the CagYΔAP OMC, the C1 ab initio 3D structure was used as the initial model for a non-uniform refinement imposing C14 symmetry. This resulting 3D volume was used to create soft masks for both particle subtraction and local refinement. After particle subtraction and refinement, a final local refinement was performed, which yielded a map with a final resolution of 3.8 Å. For the focused refinement of the CagYΔAP PR, the C1 ab initio model was used as the initial model for a non-uniform refinement imposing C17 symmetry. This resulting volume was used to create soft masks for both particle subtraction and local refinement. After particle subtraction and refinement, a final local refinement was performed, which yielded a final resolution of 6.6 Å (Fig S5).

Fig S6 shows data processing steps for cryo-EM analysis of T4SS complexes purified from the ΔcagM mutant. For analysis of the ΔcagM OMCC, 78,268 particles were selected by template picking in cryoSPARC. After iterative 2D classification, 2D classes with clear secondary structural features were retained, containing a combined 4,251 particles. These particles were subjected to a reference-free initial 3D reconstruction (ab initio model in cryoSPARC) designating two 3D classes, no applied symmetry, and a resolution range of 35–12 Å. The best 3D class contained 3,009 particles and had features with 17-fold symmetry. To attempt to visualize any OMC density in this 3D map, the volume was used in a non-uniform refinement imposing C14 symmetry. However, the C14 reconstruction contained artifacts and lacked any secondary structural features. To determine a higher resolution structure of the ΔcagM PR, the best volume was used as an initial 3D model for non-uniform refinement with C17 symmetry. This map was used to create a soft mask for local refinement. Local refinement with C17 imposed symmetry resulted in a map with 4.1 Å resolution calculated for the "tight mask" at FSC = 0.143; however, this resolution does not match the secondary features present in the map or the resolutions shown in the local resolution map (Fig S7). Taking these observations into consideration, the more accurate resolution estimation appears to be 8.5 Å as calculated by cryoSPARC when using the criterion of a "loose mask" at FSC = 0.5. This resolution matches the features seen in this map and agrees with the local resolution calculations.

Map-to-model cross-correlation values for the ASU were calculated using Phenix (Liebschner et al, 2019). Map–map correlations between mutant and WT maps were done in chimera using the "measure correlation" command. ChimeraX was used to make figures of maps and models (Pettersen et al, 2021).

## Data Availability

The cryo-EM maps have been deposited in the Electron Microscopy Data Bank under accession codes EMD-44587 (ΔcagT PR), EMD-42290 (CagYΔAP OMC), EMD-42392 (CagYΔAP PR), and EMD-42393 (ΔcagM PR).

## Supplementary Information

## Acknowledgements

We thank the TL Cover and MD Ohi laboratories for helpful discussions, and we thank Ashleigh Raczkowski, Vinson Lam, and Alexandra Rizo for cryo-EM advice. The UM Cryo-EM Facility has received generous support from the U-M Life Sciences Institute, the U-M Biosciences Initiative, and the Beckman Foundation. The Vanderbilt Institute for Infection, Immunology, and Inflammation provided support to SC Tran and KN Bryant. This work was supported by the NIH AI118932 (to TL Cover and MD Ohi), AI039657 (to TL Cover), CA116087 (to TL Cover), T32AI112541, T32GM008320, DGE 1841052 (to JR Roberts), NIH S10OD030275 (to MD Ohi), and the Department of Veterans Affairs BX004447 (to TL Cover). Mass spectrometry analysis was supported by the Vanderbilt Digestive Diseases Research Center (P30DK058404) and the Vanderbilt-Ingram Cancer Center (P30 CA068485).

### Author Contributions

JR Roberts: data curation, formal analysis, validation, investigation, visualization, and writing—original draft, review, and editing.
SC Tran: data curation, formal analysis, validation, investigation, visualization, and writing—original draft, review, and editing.
AE Frick-Cheng: validation, investigation, visualization, and writing—original draft, review, and editing.
KN Bryant: formal analysis, investigation, visualization, methodology, and writing—original draft, review, and editing.
CD Okoye: investigation, methodology, and writing—original draft, review, and editing.
WH McDonald: data curation, formal analysis, and methodology.
TL Cover: conceptualization, resources, data curation, formal analysis, supervision, funding acquisition, validation, investigation, methodology, project administration, and writing—original draft, review, and editing.
MD Ohi: conceptualization, resources, formal analysis, supervision, funding acquisition, validation, investigation, visualization, methodology, project administration, and writing—original draft, review, and editing.

### Conflict of Interest Statement

The authors declare that they have no conflict of interest.

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
