## [Reviewer comments · Life Science Alliance]

Life Science Alliance

Subdomains of the *H. pylori* Cag T4SS outer membrane core complex exhibit structural independence

Jacquelyn Roberts, Sirena Tran, Arwen Frick-Cheng, Kaeli Bryant, Chiamaka Okoye, W. McDonald, Timothy Cover, and Melanie Ohl

DOI: <https://doi.org/10.26508/lsa.202302560>

Corresponding author(s): *Melanie Ohl, University of Michigan-Ann Arbor*

Review Timeline:

Submission Date:	2023-12-26
Editorial Decision:	2024-01-26
Revision Received:	2024-03-16
Editorial Decision:	2024-03-20
Revision Received:	2024-03-28
Accepted:	2024-03-28

Transaction Report:

January 26, 2024

Re: Life Science Alliance manuscript #LSA-2023-02560-T

Dr. Melanie Ohi
University of Michigan
Nashville 37232

Dear Dr. Ohi,

Thank you for submitting your manuscript entitled "The *Helicobacter pylori* Cag Type IV secretion system periplasmic ring is stable in the absence of outer membrane cap components" to Life Science Alliance. The manuscript was assessed by expert reviewers, whose comments are appended to this letter. We invite you to submit a revised manuscript addressing the Reviewer comments.

Thank you for this interesting contribution to Life Science Alliance. We are looking forward to receiving your revised manuscript.

Sincerely,

B. MANUSCRIPT ORGANIZATION AND FORMATTING:

Reviewer #1 (Comments to the Authors (Required)):

Roberts et al here provide information regarding the organisation of the Cag T4SS, and shed light on details of how the T4SS assembly and that the periplasmic ring and the OMC is not directly coupled. The study is clear and well written, and provides new details in T4SS structure.

Overall I don't have much to remark on, but one thing that I would like to see more of in the discussion is a larger effort to put this new data in relation to other T4SSs. The authors mention this in the last paragraph, but it would strengthen the discussion if this speculation could be expanded and described in more detail. This would allow readers to put the work in more perspective.

Minor comments:

* The scale bar used in multiple figures is outside the figure and it can be unclear if it is the entire image width that's noted or just part of it for each number. I recommend making the scale bars more consistent and have them inside the figure.

* Fig 1. The resolution is really bad in this picture, with visible pixels. I'm guessing that it's something in the system, but needs to be fixed. The other figures do not have this issue.

Referee Cross-Comments:

Referee #3 raises concerns over about the cryo EM processing used in the deltacagT OMCC complex. The comments go beyond my level of expertise in SPA analysis, but it is important that this criticism is addressed by the authors.

Reviewer #2 (Comments to the Authors (Required)):

The authors have assembled a nice update on Cag subunit contributions to elaboration of the Cag OMCC, which is a large substructure of the Cag Type IV Secretion System (T4SS). Although prior work by this group showed that deletions of two subunits, CagM and CagT, disrupt assembly of the OMCC, structural perturbations were not known at a molecular level. Here, by spCryoEM, they show that deletions of either subunit abolish assembly of an outer ring of the OMCC that they have termed the OMC. This was not completely unexpected in view of prior lower-resolution structures this group has generated for these mutant machines. Remarkably, however, the deletions have no effects on assembly of the inner portion of the OMCC termed the PR. Thus, the work highlights the structural independence of the two major sub-domains of the OMCC despite the fact that both are required for machine function. Additionally, the authors report that a deletion of another sub-domain, an α -helical projection (termed the AP) of the CagY subunit that extends from the PR and presumptively forms an OM channel, has no discernable effect on the OMCC structure. As usual, the experimentation is rigorous and the findings are presented in a well-crafted manuscript. The structural advances are critical to our understanding of how this medically-important nanomachine is architecturally configured. The findings also set the stage for detailed comparisons with other T4SSs, once structures of mutant machines deleted of components of the outer or inner rings are generated. This reviewer has only a few minor comments for the authors to address.

1. Although the manuscript is clear and very well-written, there is also a fair bit of redundancy especially in the Intro and Discussion. It could be streamlined considerably.

2. L. 122. The Frick-Cheng paper nicely described structural effects of various mutations, but not functional consequences. This citation should be deleted here and substituted with the Johnson et al. paper. Having reread that paper, it remarkably reports the CagY is not required for pilus production, although CagM, CagX, and CagT are required. This might be mentioned here as well.

3. L. 72. Change to "Many conjugation systems and the" There are many conjugation systems that are 'expanded', e.g., those carried by IncF, IncH, and IncI plasmids.

4. L. 213. Add references here.

5. L. 308. While I agree with this statement in principle, the use of CagT as an example of a 'conserved' (VirB7-like) subunit isn't

really appropriate. The only conserved feature of VirB7-like subunits is their N-terminal lipid modifications, which isn't essential for function of various T4SSs. Arguably, CagT should be considered a Cag-specific subunit. Regardless, the sentence is probably accurate.

6. L. 382. Mutational analyses by Liu et al., 2022 showed that deletion of VirB9-like TraK abolished F pilus production but not substrate transfer, implying that the central cone (equivalent to I-layer) remains intact despite deletion of the outer ring complex (equivalent to O-layer or OMC). More recently, further analyses of mutant and chimeric T4SSs elaborated by F systems established that VirB7-like TraV as well as the AP associated with VirB10-like TraB are dispensable for substrate transfer although required for pilus production (bioRxiv). Together, the data agree with and expand on the findings of this study, namely, that the outer rings (OMCs or O-layers) are not critical for elaboration of functional inner cones, rings or PRs. Surprisingly, however, in the F system the outer ring is not required for substrate transfer but is required for pilus production. This shouldn't be mentioned here, it's just an interesting comparison.

7. Figs. 2 and 4. In comparing the PR structures from the Δ cagT and Δ cagM mutants, there are a couple of notable differences. In the Δ cagM structure (panel 4E, F), there are densities that project upward and there are gaps between them; this appears to be attributable to a density that is not present in the Δ cagT structure (both the points and gaps are missing in this structure (2G,H)). Can the authors explain these differences - it seems that the deletions in fact impose different structural consequences on the PR. Do the points correspond to the CagY AP domains; if so, the Δ cagT mutation seems to alter their structures, perhaps preventing formation of the AP channel which could account for the nonfunctionality of the Δ cagT mutant machines.

Reviewer #3 (Comments to the Authors (Required)):

In this manuscript Roberts et al. examine the assembly and stability of *H. pylori* Cag T4SS outer membrane core complexes (OMCC) by LC-MS and electron microscopy, building on the work from a previous study (Sheedlo et al, 2020). The authors purified OMCC complexes from strains lacking peripheral components (Δ cagT and Δ cagM) and generated cryo-EM volumes that showed weak density for the outer membrane cap (OMC) but strong, similar to wild-type density for the periplasmic ring (PR). Complexes purified from a CagY antenna projection (AP) deletion strain had no apparent effect on OMCC assembly hinting that the AP is not directly involved in OMCC assembly but corroborates its role in engaging the outer membrane (OM). Despite the rather limited novelty compared to their previous studies, the text is clear and easy to interpret, and the results are interesting and support the conclusions of the manuscript. However, there are technical concerns regarding the processing of the Δ cagT cryo-EM data and overall fits to the maps that warrant clarification.

Major criticisms

1) There appears to be a major issue with the 5.7 Å volume generated from cryo-EM data processing of the Δ cagT OMCC complex. The authors used symmetry expansion (C17) followed by additional processing (both ab initio and local refinements) in which C17 symmetry was further imposed to the symmetry-expanded particle set. This is not appropriate and results in artificially inflated GS-FSC curves arising from replicated particles. Symmetry expansion requires all downstream processing to be done without imposing further symmetry (i.e. in C1). It is also suspicious that the GS-FSC curve for this map (Supplemental Figure 2A) was calculated outside of cryoSPARC and without masks, compared to all other GS-FSC curves presented in the study, which were performed with appropriate masks within cryoSPARC. Further, the authors claim a resolution of 4.5 Å (line 464) that is not supported by the unconvincing GS-FSC curve (unmasked 5.7 Å).

2) Some of the rigid body model fits do not seem to match well with the density presented in the figures (i.e. Figure 3E,H and Figure 4F). Can this discrepancy be explained in the context of the deletion strain complexes? That is, are there some minor conformational changes occurring in the OMC/PR complexes derived from these new complexes that should be reported and compared to wild type complexes? It would help if the authors presented fit scores (i.e. cross-correlation scores between map and model, from Phenix or ChimeraX) for subunits of the asymmetric unit and for the entire multimeric assembly for all deletion complexes compared to wild type.

Minor criticisms

1) Panels in Figure 1 are very poor quality (low resolution) which makes them difficult to interpret.

2) Figure 1 legend. 6X6J is presented as the PR complex, but the PDB code for the OMC (6X6S) should also be included.

3) Figure 4A. The negative stain micrograph appears blurry compared to the other negative stain data presented.

4) CryoEM statistics table (Table S1) should also include defocus range, global map resolution by gold-standard FSC (at threshold of 0.143), local map resolution range, imposed symmetry, magnification, initial number of particles.

5) Supplemental Figure 6, lines 72-73 "...in 3D structures of the OMC and PR from the Δ cagM mutant". Only the PR is presented in this figure.

6) Supplemental Figure 6. The authors should comment on why the GS-FSC curve presented here (for Δ cagM), while at nominally higher resolution than other reconstructions presented, is extremely noisy and lacks the distinct drop-off traditionally seen for these curves (like the curves presented in Supplemental Figure 4).

7) Line 276-277: "The 3D reconstruction of the PR from the Δ cagM mutant reached 4.6 Å with 17-fold symmetry imposed". But the corresponding figure legend (lines 581-582) states "...~8 Å resolution 3D EM density map of the PR with 17-fold applied symmetry". And the corresponding GS-FSC curve (Supplemental Figure 6) reports a global resolution of 4.1 Å. Ensure resolutions are consistent across all maps presented.

8) Apart from ab initio routines presented in the study, was any 3D classification performed on the Δ cagM/ Δ cagT or the CagY-AP datasets to determine if any differences existed in the assembly or asymmetric unit, even if minor, compared to the wild-type maps?

9) Based on these results of this study can any comment be made on the proposed assembly of the OMCC? Do the results suggest that the peripheral components of the complex (Cag3, CagT, CagM) assembled after a CagXY core is formed?

Thank you for providing us with reviews of our manuscript, "The *Helicobacter pylori* Cag Type IV secretion system periplasmic ring is stable in the absence of outer membrane cap components". We appreciate the reviewers' careful reading of our work and find their comments fair, helpful, and insightful. We agree with the reviewer summaries about the strengths and limitations of this work and have updated the manuscript to address their concerns and suggestions. Our responses to the reviewers' comments are italicized. Our changes in response to reviewer comments are colored in blue and have been marked in the revised manuscript.

Reviewer #1: Roberts et al here provide information regarding the organization of the Cag T4SS, and shed light on details of how the T4SS assembly and that the periplasmic ring and the OMC is not directly coupled. The study is clear, well written, and provides new details in T4SS structure.

Overall, I don't have much to remark on, but one thing that I would like to see more of in the discussion is a larger effort to put this new data in relation to other T4SSs. The authors mention this in the last paragraph, but it would strengthen the discussion if this speculation could be expanded and described in more detail. This would allow readers to put the work in more perspective.

We appreciate the reviewer's assessment of our experimental results, and as suggested, we have added a few sentences in the discussion about potential implications of our studies in the context of other T4SSs (now page 15-16 of revised manuscript L. 373-378).

Minor comments:

* The scale bar used in multiple figures is outside the figure and it can be unclear if it is the entire image width that's noted or just part of it for each number. I recommend making the scale bars more consistent and have them inside the figure.

As suggested, we have now made the position and labeling of the scale bars consistent across all the figures and supplemental figures in the manuscript.

* Fig 1. The resolution is really bad in this picture, with visible pixels. I'm guessing that it's something in the system, but needs to be fixed. The other figures do not have this issue.

We are not sure what happened to cause Figure 1 to become pixelated. We have now saved this figure at a higher resolution. Hopefully this solves the issue.

Referee Cross-Comments:

Referee #3 raises concerns over about the cryo EM processing used in the deltagagT OMCC complex. The comments go beyond my level of expertise in SPA analysis, but it is important that this criticism is addressed by the authors.

The reviewers are correct that this is an important criticism to address. Please see our detailed response to Reviewer #3.

Reviewer #2: The authors have assembled a nice update on Cag subunit contributions to elaboration of the Cag OMCC, which is a large substructure of the Cag Type IV Secretion System (T4SS). Although prior work by this group showed that deletions of two subunits, CagM and CagT, disrupt assembly of the OMCC, structural perturbations were not known at a

molecular level. Here, by spCryoEM, they show that deletions of either subunit abolish assembly of an outer ring of the OMCC that they have termed the OMC. This was not completely unexpected in view of prior lower-resolution structures this group has generated for these mutant machines. Remarkably, however, the deletions have no effects on assembly of the inner portion of the OMCC termed the PR. Thus, the work highlights the structural independence of the two major sub-domains of the OMCC despite the fact that both are required for machine function. Additionally, the authors report that a deletion of another sub-domain, an α -helical projection (termed the AP) of the CagY subunit that extends from the PR and presumptively forms an OM channel, has no discernable effect on the OMCC structure. As usual, the experimentation is rigorous and the findings are presented in a well-crafted manuscript. The structural advances are critical to our understanding of how this medically-important nanomachine is architecturally configured. The findings also set the stage for detailed comparisons with other T4SSs, once structures of mutant machines deleted of components of the outer or inner rings are generated. This reviewer has only a few minor comments for the authors to address.

We thank the reviewer for their comments and the careful reading of our work.

1. Although the manuscript is clear and very well-written, there is also a fair bit of redundancy especially in the Intro and Discussion. It could be streamlined considerably.

As suggested, we have worked to streamline the introduction and discussion to reduce redundancy.

2. L. 122. The Frick-Cheng paper nicely described structural effects of various mutations, but not functional consequences. This citation should be deleted here and substituted with the Johnson et al. paper. Having reread that paper, it remarkably reports the CagY is not required for pilus production, although CagM, CagX, and CagT are required. This might be mentioned here as well.

*We have added the reference suggested by the reviewer, but we have retained the Frick-Cheng reference since Fig. 3A in that publication shows functional analyses (IL-8 secretion data) relevant to Cag T4SS function. Our methods for isolating the T4SS OMCC do not yield pilus components; thus, our results cannot provide any insight about relationships between the Cag T4SS OMCC and pilus production. Moreover, there are many features of the putative Cag T4SS pili that are poorly understood, making this a complicated topic. For example, the pilus-like structures were never observed in proximity with *H. pylori* Cag T4SS OMCCs in a previous cryo-ET study (Chang et al., Cell Reports 2018). For these reasons, we prefer to keep the current manuscript focused on properties of the Cag T4SS OMCC and not discuss putative Cag T4SS pili.*

- (now L. 105-107): All five proteins found in the Cag T4SS OMCC are required for Cag T4SS activity (Fischer et al., 2001; Johnson et al., 2014; Frick-Cheng et al, 2016).

3. L. 72. Change to "Many conjugation systems and the" There are many conjugation systems that are 'expanded', e.g., those carried by IncF, IncH, and IncI plasmids.

We thank the reviewer for providing this suggested change. We have reworded this sentence.

- (now L. 55-57) Now reads: *The *Agrobacterium tumefaciens* VirB/VirD4 T4SS, several conjugation systems (e.g., pKM101 and R388), and the *Xanthomonas citri* T4SS are*

considered prototypical or “minimized” T4SSs (Costa *et al*, 2021; Sheedlo *et al*, 2022; Costa *et al*, 2023).

4. L. 213. Add references here.

- *We have added the following references:*
(now L. 201-204) Another Cag T4SS protein that contacts the outer membrane is CagY (Chung *et al*, 2019; Sheedlo *et al*, 2020). The CagY AP motif is highly conserved among VirB10 family members, is predicted to form a channel in the outer membrane, and is required for *H. pylori* Cag T4SS activity (Tran *et al*, 2023).

5. L. 308. While I agree with this statement in principle, the use of CagT as an example of a 'conserved' (VirB7-like) subunit isn't really appropriate. The only conserved feature of VirB7-like subunits is their N-terminal lipid modifications, which isn't essential for function of various T4SSs. Arguably, CagT should be considered a Cag-specific subunit. Regardless, the sentence is probably accurate.

To more precisely address how CagT and VirB7 are conserved, we have changed the sentence highlighted by the reviewer to read:

- (now L. 133-135) CagT is a lipoprotein that has limited sequence similarity to VirB7 homologs, but shares structural relatedness to *X. citri* VirB7 within the N-terminal region (McClain *et al*, 2020; Fischer, 2011; Backert *et al*, 2017; Chung *et al*, 2019; Sgro *et al*, 2018).
- (now L. 292-295) Our careful examination in the current study of T4SS complexes purified from strains lacking either CagT, a Cag T4SS component whose N-terminus is structurally related to the corresponding region of *X. citri* VirB7 (Sheedlo *et al*, 2020; Sgro *et al*, 2018), or CagM, a *H. pylori*-specific protein, show that this model is too simplistic.

6. L. 382. Mutational analyses by Liu *et al*, 2022 showed that deletion of VirB9-like TraK abolished F pilus production but not substrate transfer, implying that the central cone (equivalent to I-layer) remains intact despite deletion of the outer ring complex (equivalent to O-layer or OMC). More recently, further analyses of mutant and chimeric T4SSs elaborated by F systems established that VirB7-like TraV as well as the AP associated with VirB10-like TraB are dispensable for substrate transfer although required for pilus production (bioRxiv). Together, the data agree with and expand on the findings of this study, namely, that the outer rings (OMCs or O-layers) are not critical for elaboration of functional inner cones, rings or PRs. Surprisingly, however, in the F system the outer ring is not required for substrate transfer but is required for pilus production. This needn't be mentioned here, it's just an interesting comparison.

We thank the reviewer for their interesting thoughts on how this work fits with an overall comparison of T4SS architecture. We now cite these works in the discussion:

- (now L. 375-378) Within the F-type T4SS, the OMCC is organized into a central cone (analogous to the I-layer) and an outer ring (analogous to the O-layer); therefore, VirB7 could potentially be required for stability of the central cone in this OMCC (Amin *et al*, 2021; Liu *et al*, 2022; Kishida *et al*, 2024).

7. Figs. 2 and 4. In comparing the PR structures from the Δ cagT and Δ cagM mutants, there are a couple of notable differences. In the Δ cagM structure (panel 4E, F), there are densities that project upward and there are gaps between them; this appears to be attributable to a density

that is not present in the Δ CagT structure (both the points and gaps are missing in this structure (2G,H). Can the authors explain these differences – it seems that the deletions in fact impose different structural consequences on the PR. Do the points correspond to the CagY AP domains; if so, the Δ CagT mutation seems to alter their structures, perhaps preventing formation of the AP channel which could account for the nonfunctionality of the Δ CagT mutant machines.

The reviewer is correct that there are some differences between the PR maps from Δ cagT and Δ cagM mutants. This general observation was also commented on by Reviewer 3. Although we were able to determine structures of the PRs from the OMCC purifications from the Δ cagT and Δ cagM mutants, these particles are more heterogenous than what we see when analyzing wild-type OMCCs, limiting the resolution of our maps (especially the Δ cagT PR map). For this reason, we hesitate to speculate on the significance of these differences and would rather focus emphasis on the ability of PRs to maintain their overall structural integrity and 17-fold symmetry, even when the OMC is not structured.

The reviewer also inquired whether the “points” present in the Δ cagM PR map, but not present in the Δ cagT PR map, correspond to specific CagY domains. These densities correspond to residues 1558-1603 in CagY, not the CagY AP domain (residues 1793-1863) found in the OMC. Since the OMCs were not structurally organized in either the Δ cagT or Δ cagM purifications, we do not have a biochemical (or biological) explanation for why the α -helix formed by CagY residues 1558-1603 is structured in Δ cagM but not in Δ cagT PRs.

*We have now added some of these points to an updated discussion. Additionally, Reviewer 3 suggested we calculate cross correlation scores between mutant and wild-type maps and models to provide a more quantitative comparison of these structures. These results are included in a new supplemental figure (**Supplemental Figure 3**) and are also mentioned in the results section.*

Text changes include:

- (now L. 187-190). While there are some minor differences between the high resolution wild-type PR model and the Δ cagT PR map (Supplemental Figure 3A), likely due to the heterogeneity in the Δ cagT particles, the overall structural integrity of the PR is preserved.
- (now L. 227-235) 3D single particle cryo-EM analysis of OMCCs purified from the CagY Δ AP mutant showed that although the complexes clearly lack the AP extension, the maps of the OMC and the PR (at the resolutions of 3.8 Å and 6.6 Å, respectively) are very similar to the corresponding maps of the wild-type OMC and PR (Figure 3C-H and Supplemental Figure 3B,C). CagY in the CagY Δ AP OMC, other than missing the helix-loop-helix regions (corresponding to the deleted amino acids 1793-1863), adopts the same conformation as CagY in the wild-type OMC (Figure 3C-H). The CagY Δ AP PR also has the same overall organization as the wild-type PR, although there are some minor differences that cannot be directly modeled at the current resolution of the map (Supplementary Figure 3C).
- (now L. 273-275) While there are some subtle differences between the 3D structure of the Δ cagM PR and WT PR (Supplemental Figure 3), this analysis shows that the Δ cagM PR, even without a structured OMC, retains its overall structural organization.

Reviewer #3: In this manuscript Roberts et al. examine the assembly and stability of *H. pylori* Cag T4SS outer membrane core complexes (OMCC) by LC-MS and electron microscopy, building on the work from a previous study (Sheedlo et al, 2020). The authors purified OMCC complexes from strains lacking peripheral components (Δ cagT and Δ cagM) and generated cryo-EM volumes that showed weak density for the outer membrane cap (OMC) but strong, similar to wild-type density for the periplasmic ring (PR). Complexes purified from a CagY antenna projection (AP) deletion strain had no apparent effect on OMCC assembly hinting that the AP is not directly involved in OMCC assembly but corroborates its role in engaging the outer membrane (OM). Despite the rather limited novelty compared to their previous studies, the text is clear and easy to interpret, and the results are interesting and support the conclusions of the manuscript. However, there are technical concerns regarding the processing of the Δ cagT cryo-EM data and overall fits to the maps that warrant clarification.

We thank the reviewer for the careful reading of this manuscript. We especially appreciate the reviewer's comments about the data processing steps used to determine the structure of the Δ cagT PR.

Major criticisms

1) There appears to be a major issue with the 5.7 Å volume generated from cryo-EM data processing of the Δ cagT OMCC complex. The authors used symmetry expansion (C17) followed by additional processing (both ab initio and local refinements) in which C17 symmetry was further imposed to the symmetry-expanded particle set. This is not appropriate and results in artificially inflated GS-FSC curves arising from replicated particles. Symmetry expansion requires all downstream processing to be done without imposing further symmetry (i.e. in C1). It is also suspicious that the GS-FSC curve for this map (Supplemental Figure 2A) was calculated outside of cryoSPARC and without masks, compared to all other GS-FSC curves presented in the study, which were performed with appropriate masks within cryoSPARC. Further, the authors claim a resolution of 4.5 Å (line 464) that is not supported by the unconvincing GS-FSC curve (unmasked 5.7 Å).

We agree with the reviewer's concerns about the refinement of 17-fold symmetrical Δ cagT PR and the resulting GS-FSC plot. The reviewer is correct that this structure was refined in a way that would lead to inflated resolution values via the GS-FSC curves. To address this error, we have reprocessed the data, first calculating an ab-initio model designating two classes without imposing symmetry. Both resulting ab-initio models looked similar, and one was chosen as the initial model for further refinement in cryoSPARC using local refinement, a rotation search of 10°, and C17 symmetry. This led to an EM density map with an 8.0 Å resolution.

The following changes have now been made in the manuscript:

1. Figure 2F, G, and H have been redone using the 8.0 Å 3D map processed as described above.
2. Supplemental Figures 1 and 2 have been updated to show the data processing steps, the new GS-FSC, Euler angle plot, and local resolution map.
3. The methods in the "Cryo-EM data analysis section" have been updated appropriately (now p.19 L. 452-465)
4. (now L. 184-185) "A 3D structure of the PR with 17-fold symmetry applied reached 8.0 Å resolution..."

2) Some of the rigid body model fits do not seem to match well with the density presented in the figures (i.e. Figure 3E,H and Figure 4F). Can this discrepancy be explained in the context of the deletion strain complexes? That is, are there some minor conformational changes occurring in the OMC/PR complexes derived from these new complexes that should be reported and compared to wild type complexes? It would help if the authors presented fit scores (i.e. cross-correlation scores between map and model, from Phenix or ChimeraX) for subunits of the asymmetric unit and for the entire multimeric assembly for all deletion complexes compared to wild type.

Reviewers 2 and 3 correctly note that there do appear to be some structural differences when comparing the T4SS maps calculated from complexes purified from the deletion strains. As discussed in Reviewer #2's seventh critique, due to the overall heterogeneity of the complexes purified from the mutants, we are not sure how much significance should be attributed to these minor differences and instead think the focus should be that the PRs from these mutants maintain their overall structural integrity and 17-fold symmetry, even when the OMC is not structured.

As requested by the reviewer, we calculated the cross-correlation scores between the mutant and wild-type maps and models for both the asymmetric units and the multimeric maps using Phenix. These results are included in a new supplemental figure (Supplemental Figure 3). This analysis supports the conclusion that the PR remains globally structured in complexes purified from the Δ cagM and Δ cagT mutants, even when the OMCs are not structured. While there are some differences between the various maps, it is not possible to evaluate the biochemical or biological significance of any of the differences.

Minor criticisms

1) Panels in Figure 1 are very poor quality (low resolution) which makes them difficult to interpret.

We are not sure what happened to this figure when it was uploaded to the journal. We have now saved this figure at a higher resolution. Hopefully this fixes the issue.

2) Figure 1 legend. 6X6J is presented as the PR complex, but the PDB code for the OMC (6X6S) should also be included.

We have added PDB 6X6S to the Figure 1 legend.

3) Figure 4A. The negative stain micrograph appears blurry compared to the other negative stain data presented.

We agree that this negative stain micrograph appears blurry compared to the other negative stain data presented in the manuscript. Having looked at multiple purifications from this mutant by negative stain, this "blurriness" is a consistent feature of all the T4SS purifications from the Δ cagM mutant when viewed by negative staining. This could happen for several reasons, including having higher background on the grids, how the PR from this mutant reacts to the stain, and/or staining protocol.

4) CryoEM statistics table (Table S1) should also include defocus range, global map resolution by gold-standard FSC (at threshold of 0.143), local map resolution range, imposed symmetry, magnification, initial number of particles.

We have added these values to the Cryo-EM statistics in Table S1. When examining the local resolution range for the CagY Δ AP PR, we noticed that the resolution calculated in cryoSPARC was outside the local resolution range (i.e., it was too high). This over estimation of resolution was due to a tight mask that cut off part of the map density. When the mask was fixed the resolution changed from 5.8 Å to 6.6 Å (no changes to the density map were made). The resolution for this map was updated in the manuscript and figures (L229, L481, Supplemental Table 1, and Supplemental Figures 3 and 4).

5) Supplemental Figure 6, lines 72-73 "...in 3D structures of the OMC and PR from the Δ cagM mutant". Only the PR is presented in this figure.

We have fixed this typo.

6) Supplemental Figure 6. The authors should comment on why the GS-FSC curve presented here (for Δ cagM), while at nominally higher resolution than other reconstructions presented, is extremely noisy and lacks the distinct drop-off traditionally seen for these curves (like the curves presented in Supplemental Figure 4).

The reviewer is correct that the GS-FSC curves calculated by cryoSPARC for this map (using tight, loose, and spherical masks) are very noisy and do not "drop off" (i.e., they have many "dips"). More importantly, the 4.1 Å resolution calculated for the "tight mask" at FSC = 0.143 does not match the secondary features present in the map or the resolutions shown in the local resolution map (Supplemental Figure 6C, ~4-10 Å resolution range). Taking these observations into consideration, we think the more accurate resolution estimation is 8.5 Å as calculated by cryoSPARC when using the criterion of a "loose mask" at FSC = 0.5. This resolution matches the features seen in this map and agrees with the local resolution calculations (compared to the more aggressive 4.1 Å resolution calculated using a "tight-mask" at FSC = 0.143). Importantly, this does not change our main conclusion, which is that T4SSs isolated from the Δ cagM mutant have PRs that remain structurally organized even in the absence of a structured OMC.

We have updated Supplemental Figure 6 (now Supplemental Figure 7), changed the appropriate values in cryo-EM data table (Supplemental Table 1), added to the methods section why FSC=0.5 was chosen for estimating the resolution of this structure (L.494-500), and updated the resolution values stated in the manuscript (L. 269).

7) Line 276-277: "The 3D reconstruction of the PR from the Δ cagM mutant reached 4.6 Å with 17-fold symmetry imposed". But the corresponding figure legend (lines 581-582) states "...~8 Å resolution 3D EM density map of the PR with 17-fold applied symmetry". And the corresponding GS-FSC curve (Supplemental Figure 6) reports a global resolution of 4.1 Å. Ensure resolutions are consistent across all maps presented.

As detailed above, we have now updated all references to the resolution of this map in the manuscript.

Text changes include:

- (now L. 269) The 3D reconstruction of the PR from the Δ cagM mutant reached 8.5 Å with 17-fold symmetry imposed, and secondary structural features expected at this resolution were visible in the map (Figure 4D-F).
- (L. 591-592) 8.5 Å resolution 3D EM density map of the PR with 17-fold applied symmetry.

8) Apart from ab initio routines presented in the study, was any 3D classification performed on the $\Delta\text{cagM}/\Delta\text{cagT}$ or the CagY-AP datasets to determine if any differences existed in the assembly or asymmetric unit, even if minor, compared to the wild-type maps?

We did perform 3D classification on the ΔcagM , ΔcagT , and CagY Δ AP datasets. These analyses did not identify any additional 3D classes for any of the datasets. We also used 3D variance analysis (3DVA) in cryoSPARC with the CagY Δ AP dataset, and no significant modes of variation were detected.

9) Based on these results of this study can any comment be made on the proposed assembly of the OMCC? Do the results suggest that the peripheral components of the complex (Cag3, CagT, CagM) assembled after a CagXY core is formed?

While the results presented in this manuscript do not provide any direct evidence for how Cag T4SSs assemble, they do support a key role for CagX and CagY in overall T4SS structural stability. Our results also show that, at least in the context of purified complexes, the regions of CagX and CagY found in the OMC require both CagT and CagM to maintain their structural organization. In general, our results support a model for the T4SS assembly pathway that was proposed from the cryo-ET analyses of Cag T4SSs visualized in wild-type, Δcag3 , and ΔcagT cells (Hu et al, mBio, 2019). In this model, CagX, CagY, and CagM first assemble into a central “cylinder” located between the inner and outer membranes that serves as the structural scaffold for the subsequent addition of CagT and Cag3 to form the OMC. We have added text to make this clearer in the revised discussion.

Text changes:

- (now L. 379-388) Finally, while these results do not provide any direct evidence for how Cag T4SSs assemble *in vivo*, they do support a key role for CagX and CagY in overall T4SS structural stability. At least in the context of purified complexes, the regions of CagX and CagY found in the OMC require both CagT and CagM to maintain their structural organization. In general, our results support a model for the T4SS assembly pathway that was proposed based on cryo-ET analyses of Cag T4SSs visualized in wild-type, Δcag3 , and ΔcagT bacterial cells (Hu et al, mBio, 2019). In this model, CagX, CagY, and CagM first assemble into a central “cylinder” located between the inner and outer membranes that serves as the structural scaffold for the subsequent addition of CagT and Cag3 to form the OMC. Further studies will be required to elucidate the sequence of steps required for T4SS assembly in intact bacteria.

March 20, 2024

RE: Life Science Alliance Manuscript #LSA-2023-02560-TR

Dr. Melanie Ohi
University of Michigan-Ann Arbor
Life Sciences Institute
210 Washtenaw Avenue
Ann Arbor, Michigan 48109

Dear Dr. Ohi,

Thank you for submitting your revised manuscript entitled "The *Helicobacter pylori* Cag T4SS periplasmic ring remains organized in absence of a structured OMC.". We would be happy to publish your paper in Life Science Alliance pending final revisions necessary to meet our formatting guidelines.

- please be sure that the authorship listing and order is correct
- please add a Category for your manuscript in our system
- please move your main, supplementary figure, and table legends to the main manuscript text after the references section
- there are callouts A, B, and C for Figure S3, but this figure doesn't have panels. Please correct
- please add callouts for Figures S1A-B; S2A-C; S4A-B; S5A-F; S6A-B; S7A-C and Table S1 to your main manuscript text
- we encourage you to revise the figure legend for Figure S6 such that the figure panels are introduced in alphabetical order

A. FINAL FILES:

B. MANUSCRIPT ORGANIZATION AND FORMATTING:

**Submission of a paper that does not conform to Life Science Alliance guidelines will delay the acceptance of your

manuscript.**

The license to publish form must be signed before your manuscript can be sent to production. A link to the electronic license to publish form will be available to the corresponding author only. Please take a moment to check your funder requirements.

Sincerely,

March 28, 2024

RE: Life Science Alliance Manuscript #LSA-2023-02560-TRR

Dr. Melanie D Ohi
University of Michigan-Ann Arbor
Life Sciences Institute
210 Washtenaw Avenue
Ann Arbor, Michigan 48109

Dear Dr. Ohi,

Thank you for submitting your Research Article entitled "Subdomains of the H. pylori Cag T4SS outer membrane core complex exhibit structural independence". It is a pleasure to let you know that your manuscript is now accepted for publication in Life Science Alliance. Congratulations on this interesting work.

DISTRIBUTION OF MATERIALS:

Again, congratulations on a very nice paper. I hope you found the review process to be constructive and are pleased with how the manuscript was handled editorially. We look forward to future exciting submissions from your lab.

Sincerely,
